# Ferulic Acid Stimulates Adipocyte-Specific Secretory Proteins to Regulate Adipose Homeostasis in 3T3-L1 Adipocytes

**DOI:** 10.3390/molecules26071984

**Published:** 2021-04-01

**Authors:** Palaniselvam Kuppusamy, Soundharrajan Ilavenil, In Ho Hwang, Dahye Kim, Ki Choon Choi

**Affiliations:** 1Grassland and Forage Division, National Institute of Animal Science, Rural Development Administration, Cheonan 330-801, Korea; kpalaselvamsmailbox@rediffmail.com (P.K.); arulvenil@rediffmail.com (S.I.); 2Department of Animal Science, College of Agricultural and Life Science, Chonbuk National University, Jeonju 54896, Korea; inho.hwang@jbnu.ac.kr; 3Faculty of Biotechnology, College of Applied Life Science, Jeju National University, Jeonju 63294, Korea

**Keywords:** 3T3-L1 adipocyte, obesity, ferulic acid, differentiation, lipolysis

## Abstract

Obesity has recently emerged as a public health issue facing developing countries in the world. It is caused by the accumulation of fat in adipose, characterized by insulin resistance, excessive lipid accumulation, inflammation, and oxidative stress, leading to an increase in adipokine levels. Herein, we investigated the capacity of a bioactive polyphenolic compound (ferulic acid (FA)) to control adipocyte dysfunction in 3T3-L1 adipocytes (in vitro). Key adipocyte differentiation markers, glycerol content, lipolysis-associated mRNA, and proteins were measured in experimental adipocytes. FA-treated adipocytes exhibited downregulated key adipocyte differentiation factors peroxisome proliferator-activated receptor-γ (PPAR-γ), CCAT enhancer binding-proteins—α (C/EBP-α) and its downstream targets in a time-dependent manner. The FA-treated 3T3-L1 adipocytes showed an increased release of glycerol content compared with non-treated adipocytes. Also, FA treatment significantly up-regulated the lipolysis-related factors, including *p-HSL*, and p-perilipin, and down-regulated ApoD, Sema3C, Cxcl12, Sfrp2, p-stearoyl-CoA desaturase 1 (SCD1), adiponectin, and Grk5. Also, the FA treatment showed significantly down-regulated adipokines leptin, chemerin, and irisin than the non-treated cells. The present findings indicated that FA showed significant anti-adipogenic and lipogenic activities by regulating key adipocyte factors and enzyme, enhanced lipolysis by HSL/perilipin cascade. FA is considered a potent molecule to prevent obesity and its associated metabolic changes in the future.

## 1. Introduction

Obesity is a complex and chronic disorder that has been associated with many health disorders, including cardiovascular disease, diabetes, hyperlipidemia, hypertension, cancer, osteoarthritis, and Alzheimer’s disease [1]. Among other causes of obesity, environmental factors such as food intake and lifestyle play the most important role, especially excessive energy intake leading to increase triglyceride (TG) accumulation and adipose tissue [2]. Adipose tissue growth mainly occurs in two ways: an increase in fat cell proliferation (hyperplasia) and an increase in fat cell differentiation (hypertrophy). The development of mature adipocytes from pre-adipocyte precursors is termed adipogenesis [3]. Adipogenesis is a fully controlled process involving several transcription factors that regulate cell proliferation and differentiation. During this adipogenesis, main transcription factors stimulate the lipogenic enzyme of fatty acid synthase (*FAS*) and adipocytes protein2 (aP2) [4]. 

Lipolysis is defined as the hydrolysis of triglyceride and diglyceride with the help of hormone-sensitive lipase (*HSL*) and adipose triglyceride lipase (*ATGL*), resulting in the synthesis of fatty acids and glycerol [5]. In humans, the matured adipocytes consisting of many lipid droplets (LDs) can increase in size (0.1–0.4 µm in diameter), resulting in the development of obesity. These lipid droplets are mainly composed of triglycerides and cholesterol esters, as well as some lipoproteins such as HSL, perilipin, lipase-associated enzymes, etc. [6]. Endocrine, biochemical, and nutritional factors mainly control the activity of lipolysis signaling mechanisms. In particular, the lipolytic pathway in adipose tissue is mainly regulated by lipase enzymes and adipogenesis-associated proteins such as *ATGL, HSL, visfatin, chemerin, leptin,* and other intracellular and extracellular substances. In addition, a state of adverse energy equilibrium, such as that induced by starvation, certain drugs, and intense exercise, can induce lipolysis as a result of an increase in free fatty acid (FFA) and glycerol release from adipose tissue (AT). However, dysregulation of lipolysis may have some serious metabolic consequences, including cardiac disease, cancer, etc. [7]. Therefore, researchers are paying more attention to the treatment of obesity by regulation of lipolysis. Recently, anti-obesity drugs are common and attracted a lot of interest, but they have many side effects like diarrhea, headache, and gastrointestinal tract problems. Hence, discovering natural products with no side effects for the treatment of obesity by regulating adipogenesis and lipolysis is one of the strategies to prevent and treat obesity. 

Dietary consumption of phenolic compounds has recently received more attention because of the potent functional properties of such compounds. Many phenolic compounds have been confirmed to ameliorate several manifestations of obesity-related conditions and metabolic disorders [8]. Particularly, polyphenols are major group of secondary plant metabolites and have a variety of beneficial properties, such as antimicrobial, antioxidant, anti-ageing, antitumor, antihypertensive, anti-inflammatory, and anti-obesity activities. Ferulic acid (FA) is a hydroxycinnamic acid group of phytochemical compounds found in various green vegetables, fruits, and nuts. FA has been shown to impact anti-diabetic, cardioprotective, and anti-cancer activities. However, no studies published to date on FA have shown if it possesses anti-obesity potential by regulating adipogenic homeostasis. Therefore, in this research, we demonstrate the efficacy of FA on adipogenic homeostasis mechanisms, including modulation of novel lipolysis-associated genes and adipokines, in a 3T3-L1 pre-adipocytes in vitro model. 

## 2. Materials and Methods

### 2.1. Cell Culture and Induction of Preadipocyte Differentiation 

Mouse 3T3-L1 pre-adipocytes were purchased from the American Type Culture Collection (ATCC, Rockville, MD, USA) and were cultured in Dulbecco modified Eagle medium (DMEM) supplemented with 10% fetal bovine serum (FBS) at 37 °C in a CO_2_ (5%) incubator. After 2 days confluent, the 3T3-L1 pre-adipocytes were differentiated by the addition of 0.5 mM 3-isobutyl-1-methylxanthine (IBMX), 1 µM dexamethasone, and 1 µg/mL insulin to the growth medium. Subsequently, the culture medium was switched to DMEM with 10% FBS supplemented with 1 µg/mL insulin and was freshly replaced for the next 2 days. Then, the medium was used DMEM supplemented with 10% FBS until the end of the experiment. The FA was dissolved in DMSO and then reconstituted in each medium at the working concentration. Cell growth media was also balanced with DMSO after differentiation. The FA (10 µM)-treated and control cells were harvested on day 8 after the initiation of differentiation.

### 2.2. EZ-Cytox Assay

To determine the effect of FA on cell viability, 3T3-L1 cells (1 × 10^4^/well) were grown in a 96-well cell culture plate and were incubated for 24 h in the presence of FA at various concentrations (0 to 100 μM). After 24 h, the cells were treated with 10 μL of Ez-cytox reagent at 37 °C, and the plates were incubated for 2 h in a CO_2_ incubator. Absorbance was measured at 450 nm in a SpectraMax3X reader. Percentage of cell viability was measured by defining cell viability in non-treated cells as 100%. 

### 2.3. 3T3-L1 Adipocyte Differentiation for Glyceral Release

3T3-L1 cells were seeded in a 6-well culture plate at the density of 3 × 10^4^ cells/well. The plates were incubated at cell culture conditions (5% CO_2_, 37 °C). The culture media was replaced every 48 h. After 100% was reached, cells were incubated in growth medium for another 48 h for growth arrest, and then cells were exposed to differentiation medium (DMI) containing 10% FBS in DMEM with 0.5 mM IBMX, 1 μg/mL insulin, and 1 μM for 48 h. Then, DMI was replaced by insulin medium (1 μg/mL) for another 48 h. After that, cell culture media was replaced with a normal cell culture medium until the end of the experiment [9].

### 2.4. Quantification of Glycerol Release

Glycerol concentration in the medium was measured using a glycerol assay kit (Millipore, MA, USA) in accordance with the manufacturer’s instructions. On day 8, the differentiated adipocytes were incubated with/without 10 μM FA and isoproterenol (ISO-10 μM) for 24 h. ISO was used as a positive control. Then, the culture medium was collected and used for the quantification glycerol release, i.e., 100 μL of medium were transferred to 96-well plates and 200 μL of free glycerol assay reagent was added, and the plates were incubated at room temperature for 15 min. Then, the absorbance was read at 540 nm in a SpectraMax3X reader [10]. 

### 2.5. RNA Extraction and Quantitative Real-Time PCR Analysis

The total RNA was harvested from 3T3-L1 adipocytes using Trizol RNeasy mini kit (Qiagen, Germany) according to the manufacturer’s guidelines. The harvested RNA was quantified using UVS-99 microvolume UV-Vis spectrometer-ACT gene. Then, 500 ng of total RNA was used to synthesize cDNA using an iScript cDNA synthesis kit (Bio-Rad, California, USA). An SYBR green based quantitative real-time PCR reaction was performed on an ABI 7500 PCR system (Applied Bioscience, Foster City, CA, USA). The following primer sequences were used for quantitative real-time PCR analysis: *Sfrp2*- CTC CCA AGG TGT GTG AAG C (F) and TTG GTG TCT CTG TTG ATG TAC G (R), *Sema3C*- ACA TGG AAA CCC ACT GAC AC (F) and GGG AGC ACA CTC AAG GAA AG (R), *Cxcl12*- AGA GCC AAC GTC AAG CAT C (F) and GTT GTT GTT CTT CAG CCG TG (R), *ApoD*- AAG GGT GAA GCC AAA CAG AG (F), and AGG AGT ACA CGA GGG CAT AG (R). *β-actin*- TAT GGA ATC CTG TGG CAT CC (R) and TGG TAC CAC CAG ACA GCA CT (F). The expression pattern of the desired genes was normalized with house-keeping gene *β-actin* [9]. 

### 2.6. Protein Extraction and Western Blot Analysis

After the experimental period, 3T3-L1 adipocytes were washed with phosphate-buffered saline (PBS). After being washed with PBS twice, the cells were homogenized with 1XRIPA lysis buffer (Thermo Fisher Scientific, CA, USA), which added protein and phosphates inhibitors. The extracted protein concentration was quantified using the BCA method (Thermo Fisher Scientific, CA, USA). An equal amount of protein was resolved in 4%–20% precast SDS-PAGE gel (Bio-Rad, California, USA) and transferred onto PVDF membranes with a Trans-blot Turbo transfer system (Thermo Fisher Scientific, CA, USA). Then, the membranes were blocked with a blocking solution provided in the protein chemiluminescence kit (Invitrogen, USA) for 1 h at room temperature. After blocking, membranes were washed with 0.1% TBST wash buffer and then incubated overnight at 4 °C with specific primary antibodies (Cell Signaling Technology, Danvers, MA, USA) diluted (1:1000) in 5% BSA solution. Finally, membranes were washed and incubated (1:2000) for 1 h with the anti-IgG horseradish peroxidase-conjugated secondary antibody (Cell Signaling Technology, Danvers, MA, USA) at room temperature. After washing thrice with 0.1% TBST solution, the target proteins were detected using a chemiluminescent substrate (Invitrogen, USA) with a Chemiluminescence system. β-actin was used as a house-keeping protein. The relative band intensity of the targeted proteins to β-actin in the same sample was analyzed using ImageJ software (Wayne Rasband, National Institute of Health, USA) [11]. 

### 2.7. Determination of Adipocyte Secreted Proteins from 3T3-L1 Adipocytes by ELISA

To determine the levels of adipocytes secreted proteins such as *leptin, irisin,* and *chemerin*, on day 8, 3T3-L1 cells culture medium was collected and centrifuged at 8000 rpm 4 °C for 5 min. The extracted antigens were used to quantify the specific protein level in the treated and non-treated group. ELISA assays were performed according to the manufacturer’s protocol (LifeSpan BioSciences, Inc., Seattle, WA, USA). 

### 2.8. Statistical Analysis

Quantitative data are expressed as the mean ± standard deviation (*n* = 3). Data were analyzed (one-way ANOVA and Univariate analysis included post hoc, Duncan, and descriptive analysis parameters) with on SPSS 23.0 software (SPSS software, Inc., Chicago, IL, USA). Statistically significance was considered when a *p* value is <0.05. 

## 3. Results and Discussion

Obesity can be caused by overproduction and accumulation of fat in the body. Adipocytes play a key role in storing triglycerides (lipogenesis) and releasing fatty acids (lipolysis) to maintain body weight and energy levels [12]. There is an abnormal increase in the number (hyperplasia) and size (hypertrophy) of differentiated adipocytes from pre-adipocytes, which can cause obesity by an increase in adipose tissue mass [13]. Obesity can be avoided by regulating the balance of adipogenesis and lipolysis. The inhibition of adipose tissue production and lipolysis was studied as the target mechanism in the treatment of obesity [14]. Particularly, matured adipocytes from pre-adipocytes induced by MDI showed the growth, metabolism, fat accumulation, and utilization [15]. Hence, 3T3-L1 pre-adipocytes are a widely used model for obesity and its metabolic-disorder-related study.

Recently, several studies have demonstrated the feasibility of using phenolic compounds as alternative diet therapy for obesity prevention. Phenolic compounds have been recognized in several medicinal, edible plant, and marine resources. It is considered an essential source of nutrients in the diet and plays a key role in human health. In this context, our research group has previously examined the effect of FA, which belongs to the phenolic group, on 3T3-L1 cells in order to evaluate the possible molecular mechanism of anti-adipogenic efficacy [16]. However, the mechanism underlying the FA impact on adipocyte-specific secretory proteins that regulate adipocyte homeostasis has not been investigated. In this study, we investigated the effects of FA on cell viability, adipocyte differentiation, and lipolysis in differentiated 3T3-L1 cells. Previously, we investigated the role of FA in adipocyte differentiation and lipid accumulation. This study revealed that the FA exerts strong anti-adipocyte differentiation and lipid accumulation in 3T3-L1 cells at higher concentrations. The used concentrations did not exhibit any toxic effects on cell viabilities [16]. However, in this study, we determined the cytotoxic effect of an FA dose ranges between 100–0.78 μM in 3T3-L1 pre-adipocytes. This result demonstrated that FA at used concentrations did not reveal any toxic effects on the 3T3-L1 (Figure 1A).

At the molecular level, the addition of differentiation (DMI) medium (Insulin, Dexamethasone, and IBMX) to 3T3-L1 pre-adipocytes rapidly induces key transcriptional factors C/EBP-β and C/EBPσ, which further activates the activation of nuclear hormone receptor peroxisome proliferator-activated receptor-γ (PPAR-γ) and CCAT enhancer binding-proteins-α(C/EBP-α) [17]. These factors can control the expression of adipocyte-specific genes such as fatty acid synthase (FAS), Lipoprotein lipase (LPL), and Stearoyl CoA desaturase-1 (SCD1), which are the key regulators of the fatty acid metabolism, triglyceride uptake, and lipid storage [18,19,20,21]. In the current study, we first the investigated the effect of FA at concentrations of 5 μM and 10 μM on key transcriptional factors PPAR –γ, C/EBP-α, and lipogenic enzyme FAS in adipocyte different incubation periods (Day 6, 7, and 8). We noted different impact at 5 μM and 10 μM of FA on PPAR–γ, C/EBP-α, and FAS at different treatment periods. However, FA treatment significantly downregulated PPAR–γ, C/EBP-α, and FAS expression in adipocytes on day 8 compared to control cells. It suggested that the FA treatment for 8 days significantly reduced differentiation and lipid accumulation induced by DMI in adipocytes (Figure 1B). Further, we investigated the impact of ferulic acid at different concentrations on adipokines secretion from experimental adipocytes using ELISA kits (data not given). This data suggested that 10 μM of FA treatment significantly altered leptin, irisin, and chemerin production in differentiated adipocytes compared to control cells. Based on crucial differentiation markers and secretary factors changes, we selected 10 μM of FA for further analysis.

We then investigated the lipolytic activity of FA by measuring the glycerol content in a culture medium after 24 h incubation, as shown in Figure 1C. The FA at 10 μM could stimulate lipolysis in matured adipocytes and significantly increased the glycerol release in FA treated groups as compared with non-treated group. The 3T3-L1 cells treated with isoproterenol at 10 μM showed a higher glycerol release when compared to FA treated and control group. A previous study also evidenced that the FA treated high fat mice increased the lipid excretion and maintained the lipogenic enzyme activities [22]. Our findings are accordance with Zaklos-Szyda [23], who previously reported that *Viburnum opulus* L. juice phenolics inhibit adipogenesis stimulating lipolysis and degradation of TG and release of glycerol content in mature adipocytes. Similarly, Sohle et al. [24] reported that white tea extract (2%, 20 μg/mL) significantly increased TG degradation and free glycerol content. 

Further, to confirm the mechanism underlying FA lipolytic activity in differentiated adipocytes, protein expression levels of HSL and Perilipin were studied. Our previous study indicated that the FA treatment of high-fat-diet (HFD)-induced obese mice decreased body and fat mass. In addition, it inhibits lipid droplets accumulation in adipocytes during differentiation via inhibition of key adipogenic and lipogenic transcriptional and translational factors [16]. In the current study, we focused our main aim on HSL and Perilipin changes in experimental adipocytes. Lipolysis occurs via different proteins, particularly HSL and Perilipin. It is well known that the activation of HSL by phosphorylation via PKA at several serine residues induces translocation of HSL to lipid droplets surface, which enhance lipolytic activity [25]. Lipolysis of fat cells is closely associated with Perilipin phosphorylation, which is strongly related to the surface of lipid droplets fat cell. In addition, it suggests that the non-phosphorylation status of Perilipin provided a barrier against hydrolysis by HSL and that this barrier was removed while perilipin was phosphorylated by cAMP-dependent kinase [26,27]. In the current study, we noted increased phosphorylation of Perilipin and HSL in response to FA and ISO treatments compared to control cells, resulting in an induced lipolytic process (Figure 2). This result was concurrent with quantification results of glycerol release. Perilipin is not essential for TAG storage and lipid droplet formation. However, phosphorylation of perilipin has been linked with lipid granules and is known to stimulate lipolysis and regulate stored TAG [28]. 

We then analyzed ApoD, SEMA3C, CXCL12, and sFRP2 mRNA expressions in experimental cells (Figure 3). ApoD is a secreted glycoprotein that is structurally similar to lipocalin family protein and is mainly involved in transporting lipids and cholesterol via plasma to various tissues [29]. Hummasti et al. reported that PPAR agonist GW7845 decreased apoD, whereas LXRα agonist GW3965 increased apoD level in differentiated adipocytes [30]. SEMA3C is a novel adipokine predominantly expressed in fat cells, and it is closely associated with insulin resistance and fat cells morphology. It was highly expressed in obese samples and differentiated adipocytes [31]. C-X-C chemokine ligand 12 (CXCL12) is a chemokine that is highly secreted from adipocytes when obese. In addition, the serum concentration of CXCL12 was higher with the intake of a high-fat diet and causes insulin resistance [32,33]. Sfrp2 is a family of secreted glycoprotein that has been associated with increased adiposity [34] and insulin resistance [35] (16), and serum sFRP2 is tightly related to abnormal glucose tolerance and increased insulin secretion [34]. We analyzed changes in mRNA levels of ApoD, SEMA3C, CXCL12, and sFRP2in FA treated and control cells. FA treatment downregulates mRNA expression of ApoD, SEMA3C, CXCL12, and sFRP2 compared to control cells in the current study. An elevated level of ApoD production, derived from adenovirus-mediated gene transfer, resulted in a significant reduction in plasma triglyceride level in mice [36]. However, in our current study, we noticed down-regulated ApoD mRNA expression in FA treated adipocytes. The positive aspect of FA treatment shows reduced SEMA3C, CXCL12, and sFRP2 mRNA expression in adipocytes, which are highly related to obesity and insulin resistance. 

Insulin downstream activation of Akt promotes glucose uptake into fat and muscle cells, which lower postprandial blood glucose by an enforced modification in cellular metabolism to maintain glucose status [37]. Akt is critically involved in adipocyte differentiation, overexpression of active Akt enhanced adipocyte differentiation, and lipid and glucose uptakes in adipocytes [38]. In the current study, we noted interesting results from FA-treated adipocytes that exhibited increased Akt phosphorylation at serine-473 compared to control cells (Figure 4). However, FA inhibits adipocytes differentiation and lipid accumulations. Our previous analysis exhibited FA treatment to HFD-fed mice and 3T3-L1 adipocytes inhibit body weight, fat mass, and differentiation by p38MAPK and Erk1/2 pathways. [16]. FA may play a dual role in preventing lipid accumulation and enhancing glucose uptake by different signal pathways. SCD1 is the rate-limiting enzyme involved in converting saturated fatty acids into mono-saturated fatty acids [39]. It has been increased markedly during adipocyte differentiation [40]. Reduced SCD1 is closely associated with improved insulin sensitivity and decreased bodyweight [41,42]. In the current study, we noticed a reduced level of SCD1 in response to FA and Rosiglitazone treatment compared to control cells. It confirmed that FA treatment might improve insulin sensitivity by inhibition of SCD1 and activation of Akt by its phosphorylating level. Adiponectin is exclusively secreted from adipocytes and has been overexpressed during adipogenesis [43]. Similarly, the treatment of a cell with differentiation cocktails stimulates adiponectin secretion in adipocytes, whereas FA treatment sharply reduced its level compared to control cells. Several studies have reported that the *Grk5* gene positively regulates adipogenesis and promotes lipid storage in adipocytes. Wang et al. [44] found that the decreased adipose tissue concentration observed in *Grk5* knockdown mice is due to increased lipolysis in an in vivo model. In addition, *Grk5*^−^/^−^ mice on a high-fat diet had lower body weight and white adipose tissue mass than their wild-type counterparts due to alterations in food intake and energy expenditure. Similarly, our results showed that FA-treated 3T3-L1 adipocytes exhibited a greater than the 1.5-fold decrease in *Grk5* mRNA expression compared to the control and rosiglitazone groups (*p* > 0.05). Wnt5a is a member of the wnt family, and it is classified as a non-conical Wnt protein. It plays a role in the immune response in cell culture studies [45] and promotes adipocyte differentiation in 3T3-L1 adipocytes by wnta-mediated non-conical signaling transduction. In the current study, we observed slight decreases in Wnt5a expression in FA treated adipocytes than control cells. Wnt change in FA treatment was not a significant level at *p* < 0.05 level compared to control cells. 

Finally, we investigated whether FA regulates the lipolysis-associated adipokines production in 3T3-L1 mature adipocytes. It has been reported that the fat-cell-producing adipokine of leptin effectively regulates the fat metabolism in adipocytes [46,47,48]. Leptin increased adipogenesis and lipogenesis-related key proteins and induced pro-inflammatory cytokines [49]. In this study, differentiated adipocytes treated with FA reduced secretion of leptin compared to control cells (*p* < 0.056). This indicated that FA treatment inhibits adipocyte differentiation and lipid accumulation in adipocyte via down regulation of adipogenesis and lipogenesis-associated key transcriptional factors by inhibition of leptin secretion (Figure 5A). *Irisin* is a cell membrane protein that acts in energy expenditure, glucose tolerance, and muscle hypertrophy. There was an increased irisin level in obese non-diabetic subjects, whereas this was reduced in type-II diabetic subjects [50]. We observed an increased irisin level in differentiated adipocytes; it confirmed that irisin might be closely related to adipocyte differentiation and lipid accumulation. However, FA treatment significantly reduced the irisin level in differentiated compared to control cells (Figure 5B). This suggested that FA plays a significant role in the regulation of lipid metabolism and its disorder. In addition, Xiang et al. [51] reported that short periods of muscular endurance exercise induce irisin secretion by SAT and VAT. Chemerin is an important chemokine secreted by adipocytes that plays a key role in glucose, lipid metabolism and insulin resistance, adipose tissue inflammation, and liver pathology [52]. In the current study, chemerin concentration was increased in differentiated adipocytes compared to control cells (Figure 5C), whereas the level of chemerin was reduced in response to FA treatment compared to control (*p* < 0.05). Similarly, rosiglitazone treatment decreased chemerin level in differentiated adipocytes. Inhibition of chemerin secretion by FA might be preventing obese development, insulin resistance, and hepatosteatosis [52]. 

## 4. Conclusions

The present findings demonstrated that the adipocyte treated with ferulic acid (FA) inhibited differentiation and lipid accumulation through downregulating key transcriptional factors PPARγ, C/EBPα, and key lipogenic enzyme FAS expression. Furthermore, FA’s addition to the differentiated cells stimulates the lipolytic pathway by increasing HSL and perilipin phosphorylation. FA treatment decreased mRNA of ApoD, SEMA3C, CXCL12, and sFRP2, which are closely related to adipocyte differentiation and obesity. Finally, major adipokines leptin, irisin, and chemerin secretions were inhibited by FA treatment. Overall data suggested that the FA can regulate the lipid metabolism in adipocyte by different aspects that could be considered a potential molecule to control obesity and its metabolic disorder. However, an in-depth analysis should be needed for warrant use in future.

## Figures and Tables

**Figure 1 molecules-26-01984-f001:**
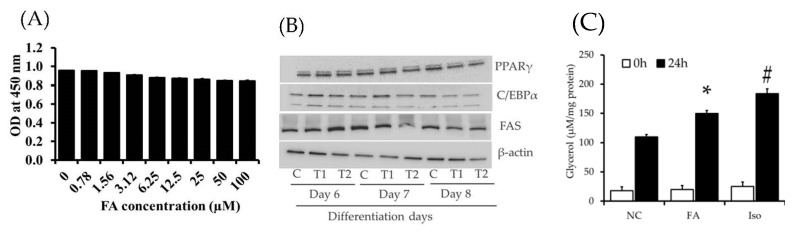
Impact of ferulic acid (FA) on cell viability, key adipocyte factors, and glycerol release. (**A**) Cells treated with different concentrations of FA and determined its toxic effect on 3T3-L1 pre-adipocytes using EZ-cytox kit; (**B**) changes in key differentiation protein markers and its downstream target in response to FA treatment at different days; (**C**) 3T3-L1 adipocytes were treated with FA and Isoproterenol (ISO) at 10 μM, followed by the amount of glycerol content release into media from differentiated adipocytes for 0 and 24 h were quantified. C: control; T1: 5 μM; T1: 10 μM; Data are represented as the mean ± standard error of the mean three replicates. * *p* < 0.05 control vs. FA treatment; # *p* < 0.05; ISO vs. control and FA treatment.

**Figure 2 molecules-26-01984-f002:**
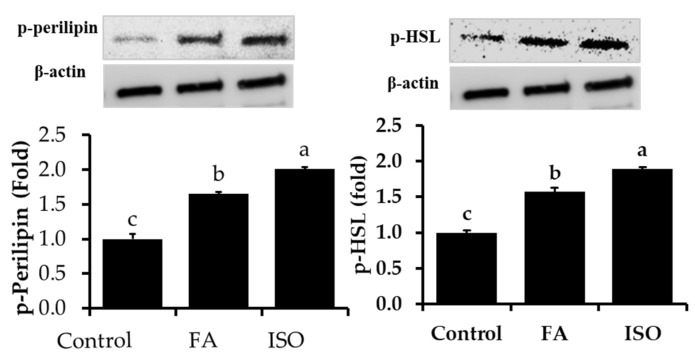
Phosphorylation levels of perilipin and hormone-sensitive lipase (HSL) were determined in experimental 3T3-L1 adipocytes with or without FA and ISO. The results are represented as the mean ± standard error of the mean three replicates. ^a,b,c^
*p* < 0.05 statistically significant difference between control and treated groups.

**Figure 3 molecules-26-01984-f003:**
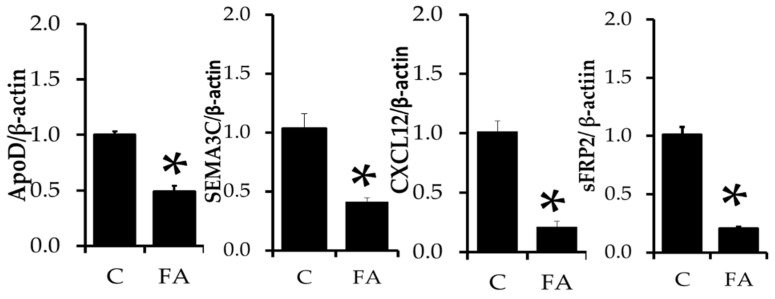
Transcriptional changes in apolipoprotein D (ApoD), semaphorin 3C (SEMA3C), C-X-C chemokine ligand 12 (CXCL12) and secreted frizzled related protein 2 (sFRP2) in FA-treated and control adipocytes. The results are represented as the mean ± standard error of the mean of three replicates. * *p* < 0.05 statistically significant difference between control and treated group.

**Figure 4 molecules-26-01984-f004:**
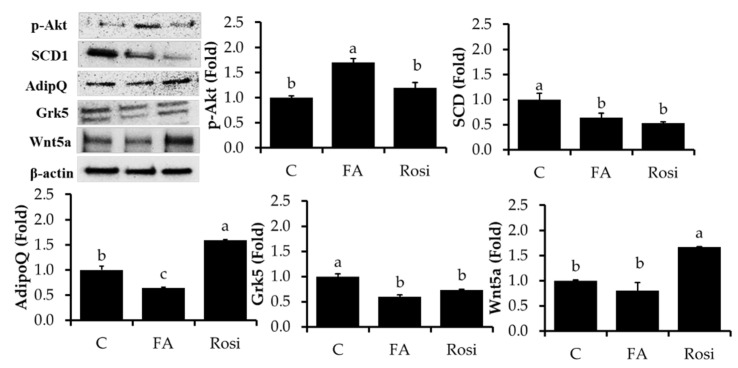
Lipolysis-associated proteins were analyzed by immunoblotting. After incubation of 3T3-L1 adipocytes with/without FA and rosiglitazone (Rosi) at 10 μM treatment for 8 days, the expression levels of *p*-perilipin, *p*-HSL (Ser 565), *p*-AKT, SCD1, Wnt5a, adiponectin Grk5, and β-actin were measured. The results are represented as the mean ± standard error of the mean three replicates. The Fluorescence intensity of the targeted proteins was normalized with housekeeping protein (β-actin) using ImageJ software. ^a,b,c^
*p* < 0.05 statistically significant differences between control and treated groups.

**Figure 5 molecules-26-01984-f005:**
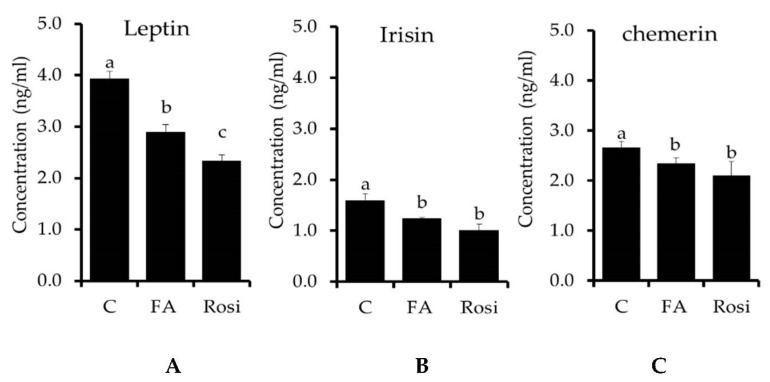
The efficacy of a ferulic acid and Rosi treatment on adipokines secretion in 3T3-L1 adipocytes on day 8, as assessed with a commercial ELISA kit. The concentrations of leptin, irisin, and chemerin secretory proteins in the cell-free media of 3T3-L1 cells treated with/without FA were measured. (**A**) Leptin concentration in experimental cells; (**B**) Irisin concentration in experimental cells; (**C**) Chemerin concentration in experimental cells. The results are represented as the mean ± standard error of the mean three replicates. ^a,b,c^
*p* < 0.05 statistically significant difference between control and treated groups.

## Data Availability

The data presented in this study are available in the article.

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
