# Peer review of "Ferulic Acid Stimulates Adipocyte-Specific Secretory Proteins to Regulate Adipose Homeostasis in 3T3-L1 Adipocytes"

_molecules, 2021, doi:10.3390/molecules26071984_

Round 1
Reviewer 1 Report
The Authors mostly answered to my previous suggestions. Additionally, the “Results and Discussion” section has been rewritten.
Still, in the main text there is lack of explanation of usage 5 and 10μM of FA, whereas the Authors wrote about lack of the ferulic acid cytotoxicity at concentration range 0-2 mM. Maybe additional figure showing the influence of FA at higher concentrations on adipocyte differentiation can be shown or direct correlation with FA doses used for in vivo studies.
Please explain why the Figures 1A and 1B are separated within the text and not numbered as Figure 1, Figure 2, etc.
Whereas the article is easy to understand, it needs English checking.
Author Response
We thank the reviewers for their critical and judicious evaluation of our manuscript and providing constructive suggestions for improving the quality and presentation of the manuscript. We have carefully considered the comments of the reviewers and revised the manuscript thoroughly considering all the points. Pointwise response to the reviewer's comments is given below.
- The Authors mostly answered to my previous suggestions. Additionally, the “Results and Discussion” section has been rewritten.
Thank for your valuable suggestion, we have modified result and discussion section in certain places of the manuscript. Also whole conclusion has been modified according to reviewer comments. All changes were made in red colored fonts.
- Still, in the main text there is lack of explanation of usage 5 and 10μM of FA, whereas the Authors wrote about lack of the ferulic acid cytotoxicity at concentration range 0-2 mM. Maybe additional figure showing the influence of FA at higher concentrations on adipocyte differentiation can be shown or direct correlation with FA doses used for in vivo studies
Yes we agreed with reviewer comment; 0.2mM FA experiment was performed and published previously. However, again we performed cytotoxic effect of FA at the different concentrations range from 0.78-100uM and included this data also in the revised manuscript. This data revealed that used concentrations of FA did not exert negative effects significantly on 3T3-L1 cells. Please see the new figure 1A
For usage 5 and 10μM of FA: We noted different impact at 5 μM and 10 μM of FA on PPAR –γ, C/EBP- α and FAS at different treatment periods. However, FA treatment significantly downregulated PPAR –γ, C/EBP- α and FAS expression in adipocytes on day 8 compared to control cells. It suggested that the FA treatment for 8days significantly reduced differentiation and lipid accumulation induced by DMI in adipocytes (Fig. 1A). Further, we investigated the impact of ferulic acid at different concentrations on adipokines secretion from experimental adipocytes using ELISA kits (Data not given). This data suggested that 10 μM of FA treatment significantly altered leptin, irisin and chemerin production in differentiated adipocytes compared to control cells. Based on crucial differentiation markers and secretary factors changes, we selected 10 μM of FA for further analysis.
- Please explain why the Figures 1A and 1B are separated within the text and not numbered as Figure 1, Figure 2, etc. Whereas the article is easy to understand, it needs English checking.
Yes we agreed with reviewer comment; now we have modified all figures and fixed in one place for easy understanding purpose (Figure 1A-C). Language of manuscript also carefully checked and changes were made in red colored fonts for easy identification
Reviewer 2 Report
I have no further comments
Author Response
Thank for your positive comments on our manuscript and Language of the manuscript has been checked carefully by renewed professors and online tool. All changes were made in red colored fonts.